# Spatial and Sectoral Determinants of Productivity: An Empirical Approach Using an Entropy Lens

**DOI:** 10.3390/e22111271

**Published:** 2020-11-09

**Authors:** Sónia de Brito, João Leitão

**Affiliations:** 1Research Center in Business Sciences (NECE), University of Beira Interior, 6200-209 Covilhã, Portugal; 2Department of Management and Economics, Faculty of Social Sciences and Humanities, Research Center in Business Sciences (NECE), University of Beira Interior, 6200-209 Covilhã, Portugal; jleitao@ubi.pt; 3CEG-IST—Centre for Management Studies of Instituto Superior Técnico, University of Lisbon, 1049-001 Lisbon, Portugal; 4ICS—Instituto de Ciências Sociais, University of Lisbon, 1049-001 Lisbon, Portugal

**Keywords:** concentration, diversification, entropy, specialization

## Abstract

This study analyzes the productive structure of Portugal in the period 2013–2017, using indicators of localization and specialization applied to 308 Portuguese local authorities. From an empirical approach using a threshold model, the following indicators are used: (i) localization quotient; (ii) specialization coefficient; (iii) Theil entropy index; (iv) rate of industrialization; and (v) the density of establishments by business size. The selected period 2013–2017 is due to the available data concerning firms located per local authority, and the choice of threshold model is justified through the possibility of assessing the non-linear effects of specialization and diversification on productivity, considering, in simultaneous terms, different regimes per business size. Estimation of the threshold model identified a positive, statistically significant relation between industrialization and productivity. Similarly, the terms of interaction between exports and diversification, and between the former and higher education institutions, shows a catalyzing effect of productivity. In addition, the most specialized micro-firms affect productivity significantly and positively, while the least specialized have the opposite effect. Small, less specialized companies have a significant and negative effect on productivity, contrasting with less specialized, medium-sized companies, which affect productivity positively. For large firms, the impact on productivity is negative for both high and low levels of specialization, reinforcing the need to fill existing gaps in strategic diversification, as well as the vertical and horizontal integration of the activities of production chains with high value added.

## 1. Introduction

The relation between spatial concentration and sector specialization, which began to be studied in the field of regional science, in recent years has emerged in the field of entrepreneurship and innovation, with it being a well-established fact that business undertakings are susceptible to geographical concentration [1,2,3], together with clear evidence that entrepreneurial activity varies considerably between countries and regions [4,5] and this phenomenon is shown to be persistent over time [6,7]. According to Aiginger and Rossi-Hansberg [8], spatial concentration and sector specialization have been studied as intrinsically related economic phenomena. Therefore, most empirical studies deal with both processes as parallels, meaning that concentration dynamics are accompanied by the same specialization dynamics. This being so, it is necessary to present the difference between spatial concentration and sector specialization, since there are always ambiguities arising from the fact that concentration is sometimes presented as equivalent to specialization. Spatial concentration is the extent to which in one country a given industry or sector is concentrated in a limited number of regions; sector specialization is the extent to which a country concentrates its industrial activity in a limited number of sectors, so that a region has a profile of a highly specialized production structure when regional production is distributed mainly over a small number of sectors [9]. In turn, unlike sector specialization, Chowdhury et al. [10] refer to sector diversification as corresponding to a concentration of production activities not in a small number of sectors but in diverse sectors.

In the literature on this topic, the concept of industrial district formulated by Marshall [11] and Becattini [12] as well as the concept of cluster popularized by Porter [13] are predominant. In addition, industrialization, representing the process by which industrial sectors come to play a dominant role in a national economy, is also closely related to phenomena of spatial agglomeration and concentration. For Chenery et al. [14], the most widespread characteristic of industrialization is that corresponding to transformation of the production structure, whereby industrial sectors typically grow more quickly than agriculture. Fujita et al. [15] mention that one essential characteristic of industrialization is spatial concentration, and that indeed industrialization is frequently accompanied by the spatial agglomeration of industrial activities.

Industrialization greatly improved productivity [16], which has a relevant role in determining a country’s economic well-being [17]. According to the OECD [18], there are different ways to measure productivity and the choice depends on the purpose of measuring it, and in many cases, the data available. Among alternative measures of productivity, such as multi-factor productivity or capital productivity, work productivity is particularly important in the economic and statistical analysis of a country. Work productivity equals the ratio between a measure of the output volume, in this case gross value added (GVA), and a measure of the use of inputs, in this case, the population employed. Therefore, the measure of the output volume reflects the goods and services produced by the workforce, while the measure of the use of inputs reflects the workforce’s time, effort, and skills.

Aiming to advance knowledge about the relation between industrialization and productivity, using the Sales Index database, this study uses indicators of localization and specialization applied to all 308 local authorities in Portugal, intending to make a generic analysis of the dynamics of their production structure in the period 2013–2017. More specifically, a threshold model is estimated, in order to test the effect of industrialization on productivity as well as other research hypotheses arising from the literature review.

In this vein, the current study uses a concept of entropy operationalized through the Theil Index, developing an economics approach focused on the analysis of both spatial and sectoral determinants of productivity. This approach aims to assess the non-linear effects of specialization and diversification on productivity, considering, in simultaneous terms, different regimes per business size. Toward this empirical application, we contribute to advancing the existent knowledge on determinants of productivity, using an Entropy index and also unveiling distinct signals and significances of the determinants studied in different specialization regimes.

To achieve the objectives proposed, the following sections present firstly a literature review, originating in industrial districts and moving toward the cluster approach, and resulting in the research hypotheses. This is followed by the empirical approach, namely the methodology, and presentation, analysis, and discussion of the results. The study ends with the conclusion, limitations, and implications.

## 2. Literature Review

### 2.1. From Industrial Districts to Clusters

The concept of the industrial district was originally presented in the *Principles of Economics* by Marshall [11], proposing that the geographical agglomeration of companies in the same or similar branches of industry leads to organizational growth and development, allowing firms to obtain economic advantages expressed by external economies. From the pioneering work by Marshall [11], and after lying dormant for decades, the concept of industrial district was only taken up again in the 1970s, in Italy, by Becattini [19], who defined it as a socio-territorial entity characterized by the active co-presence, in a limited, natural, and historically determined territorial area of a community of people and a population of industrial companies, differentiated from the traditional economic region by having industry as the dominant activity. One of the main characteristics of industrial districts is firms’ flexible specialization, i.e., the social division of work between firms, based on tasks and their interconnections [20].

Both Marshall [11] and Becattini [12,21] refer to the division of the production process among firms, but this is not viable for all products. For the social division of work among firms to be viable, it must be possible to decompose this process in terms of space and time [22]. For Becattini [21], specialization of the local workforce has the nature of a public good and is a key factor for the district’s productivity and competitiveness. Homogeneity among production units allows a great mobility of workers between companies. Another aspect recognized by Becattini [21] in industrial districts, and also mentioned earlier by Marshall [11], is that knowledge is spread both formally, referring to the teaching process or learning at work, and informally, referring to personal contacts among the various agents.

Other concepts have emerged in the literature, considering the spatial agglomeration of companies and its relation with other variables such as innovation. Highlighted here are the concept of innovative milieux, innovative systems, and the concept of learning regions. Developed by the *Groupe de Recherche Européen sur les Milieux Innovateurs* (GREMI), the concept of innovative milieux corresponds to how a company is regarded not as an isolated agent of innovation but as being inserted in a milieu with innovation potential [23]. Authors such as Aydalot [24], Ratti [25], Camagni [26,27], and Camagni and Maillat [28] state that the concept of innovative milieu has explored networks between innovation activities and space. Making a comparison between countries, states, and metropolitan areas, Jaffe et al. [29] argue that knowledge spillovers are geographically located and concentrated, meaning that the smaller the geographical area, the more significant the location and incidence of spillovers. A city can also be a rich context for the development of networks, and supporting this, Capello [30] concludes that cities that are not too large facilitate environmental balance, efficient mobility, and the possibility of maintaining a sense of belonging in the population. However, the city concept does not have the same characteristics as the notion of an innovative milieu [31,32]. In the perspective of Maennig and Ölschläger [32], if there is exchange and interaction between the city and the milieu, first of all, the whole city forms the physical basis and the milieu is formed through urban relational capital and collective learning processes, and secondly, a single specialized industry in a city forms a milieu.

Regarding the concept of innovation systems, this emerges in the literature with a focus on the national level of analysis, namely through building the theory of national innovation systems (NIS). The oldest versions of the NIS concept go back to Freeman [33], Nelson [34], and then Edquist [35], with the concept of innovation systems seeking to contemplate various factors determining the innovative process, based on the systemic nature of innovation. In the innovation system, innovation is systemic, multi-functional, and inter-organizational, being related to industrial dynamics and the relations between innovative firms and their milieu. Furthermore, at the national level, there are different possibilities for the organization of markets. In this connection, Lundvall [36] mentions that the interaction between universities, the types of interaction cultivated between specialists, and financial markets, which were analyzed separately in the literature, have gradually been considered and inserted in the perspective of systems. In recent decades, the regional issue has gained relevance due to the problem of asymmetric development and regional divergence. The accelerated globalization process and technological progress have clarified the need to deal with the matter of innovation in regions, and in this connection, it was Cooke [37] who introduced the concept of regional innovation systems (RIS), which is widely used in studies about innovation processes in regional economies [38,39,40,41,42].

An RIS can be defined as a system in which companies and other organizations are systematically concentrated in interactive learning through an institutional means characterized by immersion [40]. Added to this is the concept of learning regions developed by Cooke [43], Morgan [44], and Asheim [45], which can be considered as an attempt to synthesize the spatial models of innovation, but by highlighting the importance of the role played by institutions in regional development, it ends up being distinguishable from the other spatial models of innovation [44].

More recently, Porter [13], inspired by the work already mentioned by Marshall [11] and Becattini [19] to explain the nature of the competitiveness in industrialized countries, introduced the concept of clusters, defining them as geographical concentrations of inter-related firms, specialized suppliers, service companies, firms in related industries, and cultural and teaching institutions—for example, universities, agencies and business associations—in a given area, that promote simultaneously cooperation and competition: (i) cooperation between related firms and local institutions; and (ii) competition between rival firms, in terms of attracting and holding on to customers. Clusters correspond to the solid set of related firms located in a small geographical area, which are sometimes centered on a country’s scientific basis [46]. According to Porter [47], clusters have an important role in companies’ competitiveness, above all through the increased productivity of companies and industries, through increased innovation capacity, and through the intermediary of promoting new businesses that support innovation and give clusters scale. Firms’ productivity is increased through access to specific production factors and a specialized workforce, information, complementarities, institutions, public goods, and performance incentives. These factors bring about advantages such as a larger qualified workforce, increased specialization among suppliers, access to global markets, and reduced costs [47]. Studies have been made to define the context in which firms operate, namely using models reflected in the industrial concentration described by Krugman [48,49] and Fujita et al. [15], showing the advantages and success stories in various countries [50], as regards economic development [51] and learning processes [52,53]. Therefore, industrial concentrations result in growth through the results and advantages arising from spatial proximity [54], as is the case of the effect of the production function associated with transport costs, increasing productivity with a fixed number of production factors [55].

Clusters present benefits in the form of increased capacity for innovation and learning, technological externalities and increased flexibility and effectiveness of production and distribution systems [56], vertically disintegrated sub-contracting relations between firms specialized in different phases of production and interaction between small firms [20], local production networks [50], and interdependence [57] and firm networks that facilitate imitation and improvement [58], contributing to the development of competitive advantages for the firms located in these clusters [13,59] with a view toward cooperation rather than competition [60]. In a context of inter-connections, some researchers emphasize the importance of local learning [52,53], while others give importance to the links between market processes and institutional and cultural factors [61,62]. The literature shows that strong cooperation networks between firms and support agencies within clusters are characteristics of successful clusters [61,63], presenting differences in the importance of cooperation and competition in their environment [64].

### 2.2. Spatial Concentration and Sector Specialization: Research Hypotheses and Conceptual Model

Rodrick [65] indicates explicitly that the transition to modern industrial activities acts as a driver of growth, arguing that structural transformation is the only explanation of growth in a rapidly developing world. Later, Rodrick [66] also reveals that industry is the only sector of the economy that achieves unconditional convergence in productivity. The study by McCausland and Theodossiou [67] confirms the positive impact of industrialization on growth, also underlining that the role of the service sector in determining economic growth is not comparable to that of the industrial sector. Kathuria and Natarajan [68] analyze the determinant factors of regional growth, concluding that more industrialized regions grow more quickly. Güçlü [69] also finds evidence that the industrial sector has a positive impact on economic growth. Szirmai and Verspagen [70] assess the impact of the industrial sector on economic growth and find it has a moderately positive effect, not finding the same effect for the service sector. A study by Haraguchi et al. [71], in the context of developed and developing countries, revealed that growth stimulated by industrialization is still powerful for developing countries, despite recent allegations of reduced industrial development and the reduced relevance of industry for economic development and structural change in the economy.

More recently, Zhao and Tang [72] examined the sources of economic growth in China compared to Russia in the period between 1995 and 2008, finding that increased economic growth in Russia was stimulated largely by the service sector, which was followed by the primary sector. On the contrary, in China, increased economic growth was largely achieved through the contribution of the industrial sector and to a lesser extent by the service sector. In addition, the hypothesis of a non-linear relation between industrialization and economic growth is not rejected, according to the evidence found in the study by Ortiz et al. [73], who argue that every society should strive to achieve a minimum level of industrial technological integration before being able to reap the benefits of industrialization in the form of economic growth. If the hypothesis was rejected, it would imply that countries enjoy the benefits of industrialization in economic growth once they go beyond a certain threshold of technological integration in the industrial sector. Although the empirical evidence reflects the impact of industrialization on economic growth and not directly on productivity, with the proviso that productivity can be an indicator revealing various economic indicators, in that it provides a measure of economic growth, the above results in the following research hypotheses:

**Hypothesis** **1.**
*Industrialization is positively related to productivity.*


**Hypothesis** **1.**
*Increased industrialization has a non-linear relation with productivity.*


The impact of agglomeration on productivity can be seen according to two theories that best explain specialization and diversification, these being Marshall-Arrow-Romer (MAR) theory and Jacobs theory. Beginning with the theory of Marshall [11], Arrow [74] and Romer [75], formalized by Glaeser et al. [76] as Marshall–Arrow–Romer (MAR), which predominates in specialized environments and defends that the concentration of an industry in one region promotes knowledge spillovers between companies and facilitates innovation in a specific industry within a region. According to Saxenian [50], specialization stimulates the transmission and exchange of knowledge, ideas, and information, whether tacit or coded, about products and processes through imitations, commercial interactions, and qualified workers’ circulation among companies, without monetary transactions. However, knowledge externalities among companies only occur between firms in the same or similar industries, and so, they can only be supported by regional concentrations of the same or similar industries. Consequently, it is also assumed there can be no knowledge spillovers between industries. Frenken et al. [77] mention that MAR externalities tend to emerge when the industry the company’s main activity belongs to is relatively large. Mukkala [78] argues that workers are consequently better protected from business uncertainty and demand shocks if located in a region with a major local base of their own industry. Glaeser et al. [76] conclude that a local monopoly is better for growth than local competition, since a local monopoly restricts the flow of ideas to others, and therefore allows externalities to be internalized by the innovator. Those spillovers of an intra-industrial nature are known as externalities of location or specialization or MAR.

The theory of Jacobs [79], which prevails in diversified environments, proposes that the most important sources of knowledge spillovers are outside the industry in which a given firm operates. As the diversity of these sources of knowledge is greater in cities, Jacobs [79] also concludes that cities themselves are a source of innovation. This theory emphasizes that the variety of industries in a given geographical region promotes knowledge externalities, and consequently, innovative activity and economic growth. Furthermore, a more diversified business community in close proximity promotes opportunities to imitate, share, and recombine ideas and practices in all sectors. For Harrison et al. [80], a more diversified economy favors the exchange of necessary skills for the emergence of areas of economic activity. In this connection, for Combes [81], this assumes that technologically related sectors can come to be incorporated in the production activities of other industries. Moreover, transport and communication infrastructure that works well, proximity to markets, and better access to specialized services are additional sources of Jacobs externalities that Jacobs argues facilitate firms’ operations. Jacobs [79] uses the example of Manchester as a city specialized in textiles that failed, in contrast to the success of Birmingham, which was structurally diversified to argue that the diversification of industries in the same place, and not specialization, can promote knowledge-related externalities and lead to innovation and economic growth. Therefore, it is indicated that a diversified local production structure originates diversification externalities.

In the empirical literature, the results obtained by De Lucio et al. [82] show that MAR externalities affect productivity growth, the same not occurring with regard to Jacobs externalities. The same authors defend that below a certain threshold of specialization, MAR externalities have a negative effect on growth, and above that threshold, the opposite is true, i.e., greater specialization is better for productivity growth. Frenken et al. [77] do not find evidence of the effects of specialization on productivity, and their measurement of diversification reveals a negative impact on productivity growth, despite causing a strongly positive impact on employment growth. Mukkala [78] and Almeida [83] find evidence of specialization externalities in productivity. Beardsell and Henderson [84], Black and Henderson [85], and Henderson [86], using data on productivity, conclude that firms benefit from a more specialized industrial environment, thereby rejecting the theory of Jacobs. Dekle [87] compares the effect of MAR and Jacobs externalities on the growth of total factor productivity and employment growth and finds evidence of MAR in the former but not in the latter. Cingano and Schivardi [88] also find evidence of MAR externalities in the growth of total factor productivity but not in employment growth. None of these studies found that Jacobs externalities influence productivity growth, and Capello [89] and Henderson et al. [90] obtained similar results. Capello [89] separates large and small firms and reveals that economies of specialization have a positive impact on small firms’ productivity. Henderson et al. [90] reveal that productivity increases in high-tech sectors when there is a greater concentration of the sector. Forni and Paba [91] give support to both MAR and Jacobs externalities when they analyze empirically the effects of specialization and industrial diversification on the growth of Italian industry, arguing that the effect of industrial agglomeration is vital in regional industrial growth, also concluding that industrial specialization and diversification have a significant facilitating function for most industries. Simonen et al. [92] indicate that both moderate specialization and diversification have a positive role in regional economic growth, despite being subject to the influence of the scale of the city, the agglomeration structure, and other conditions. Yuan et al. [93] show that MAR externalities increase technical efficiency, reducing pure technical efficiency and accelerating technological progress, while Jacobs externalities increase scale efficiency and technological progress, despite contributing to diminished pure technical efficiency. According to Groot et al. [94], a more recent view of the role of MAR externalities is based on the concepts of related and non-related industries. This vision shares the idea of the positive effect of inter-sector spillovers of the Jacobs type. However, the difference lies in the fact of even knowledge spillovers being linked and flowing geographically between sectors, with the effect on growth depending on the extent to which knowledge flows through complementary or non-complementary sectors. A region specialized in a particular composition of complementary sectors will experience higher rates of growth than one specialized in sectors that do not complement each other [95]. According to this point of view, results provided by Greunz [96], Bochma et al. [97,98], and Cainelli et al. [99] indicate that companies and start-ups should agglomerate in regions where there is close technological proximity between firms. Concerning diversification, Glaeser et al. [76] argue that a local industry prospers if it faces a diversified surrounding economic structure. The results found by Batisse [100] when studying the relation between the local economic structure and the growth of Chinese provinces show that specialization has a strong negative impact on growth, whereas a more diversified industrial community has a positive impact. Capello [89] studies small and large companies, revealing that diversification externalities are more advantageous for large ones. Frenken et al. [95] assess whether the diversification of related or non-related industries favors stability and regional growth, finding that the related diversification of industries contributes to increasing employment. Considering the above, the following research hypothesis is formulated:

**Hypothesis** **3.**
*Diversification is positively related to productivity.*


According to Aw and Hwang [101], there is consensus regarding the primary role of exports in determining high levels of growth in production and productivity. Aw and Hwang [101], Bernard and Wagner [102], Bernard and Jensen [103], Aw et al. [104] and Delgado et al. [105] analyze empirically how exports and productivity are related to company structure, with the evidence revealing that exporting firms perform better than non-exporting ones, not only in terms of survival, salaries, capital intensity, and technological sophistication, but also in productivity. Now, the question raised here is if exports have a positive moderating effect on the relation between diversification and productivity. In this connection, according to Jacobs [79], in the case of a country, city, or region, these grow through a process of gradual diversification and differentiation of their economy, being stimulated by production oriented to the external market and by work efforts directed to exports. During the process of economic growth, through adding new work to the economy, it is essential that internal products come to be exported and that new products are created, for both the internal and external markets.

Returning to the vision of Jacobs [79], if a serious problem arises in the economy, this can only be solved by adding new goods and services. Considering the multiplying effect of exports, specialization of the internal production of certain goods and services for local consumption allows the latter to be exported, as the greater the specialization, the easier it becomes to export the goods, which in turn creates wealth, stimulates local employment, and makes increased imports viable.

The capacity to develop new goods and services for export is essential in this growth process, as in the same line of argument as Jacobs [79], the capacity to develop new goods and services for export is essential for the process of strengthening productivity, in that generating new exports gives room for the local expansion of work, due to the multiplying effect of exports, and puts pressure on the increased efficiency of the productive structure. In this connection, Prebisch [106] argues that diversification of the productive structure also benefits economic growth, in that it can make the country less dependent on more sophisticated imports and can therefore contribute to reducing the external imbalance and to combating low levels of economic growth. Moreover, diversification of the productive structure could lead to diversifying the export structure, reducing the dependence on income from exporting few goods, normally commodities. Imbs and Waczarg [107] consider that structural change responds basically to the commercial policy followed and economic growth, which agrees with the line taken by Chenery et al. [14], who indicate that economies that follow growth strategies guided by exports industrialize earlier, register higher rates of total factor productivity and are faster to reach the productive structure of an advanced economy. Therefore, the following research hypothesis is considered:

**Hypothesis** **4.**
*Exports have a moderating effect between diversification and productivity.*


Considering the literature review and the research hypotheses formulated, the operational model presented in Figure 1 is proposed.

## 3. Methodology

### 3.1. Concentration and Specialization: Indicators and Metrics

For Delgado and Godinho [108], the indicators of localization and specialization are measures of a descriptive nature that can characterize the production structure of each region, aiming to analyze the degree of geographical concentration/dispersion and the degree of specialization or diversification. According to Paiva [109], in calculating these indicators, the variable used should be the one ensuring the least possibility of bias in the results and also presenting the greatest possible number of sub-sectors, as the greater the sectoral disaggregation, the better the identification of regional specialization. In this context, the variable most commonly used in the literature, particularly in the classic studies by Isard [110] and later in the study by Dion [111], is the one corresponding to the number of employees by sector, and for that reason, this variable is used here. After defining the variable to be used, the sectors of economic activity considered here are primary, secondary, and tertiary. Based on the Portuguese Classification of Economic Activities, 3rd review, abbreviated to CAE-Rev.3 (Table 1), the primary sector is considered to include sections A and B of CAE-Rev.3; the secondary sector covers sections C, D, and E of CAE-Rev.3; and the tertiary sector includes sections F, G, H, I, J, L, M, N, O, P, Q, R, and S of CAE-Rev.3. Sections K, O, T, and U are not considered due to the lack of available information.

After choosing the variable to be used and the area of analysis for calculation of the indicators, the following coding is defined: *x* represents employment; *i* represents each sector of activity; *I* represents the set of sectors in an economy; *r* represents each of the local authorities in which the area of analysis is sub-divided; *R* represents the set of local authorities according to NUTS II, i.e., Algarve, Alentejo, Metropolitan Area of Lisbon, Centre, North, Autonomous Regions of the Azores and Madeira; *j* represents manufacturing industry; *PE* represents the employed population; *A* represents area in Km^2^; *E ≤ 9* represents the number of establishments with no more than nine employees; *E* ≤ 49 the number of establishments with no more than 49 employees; *E* ≤ 249 the number of establishments with no more than 249 employees; and *E* ≥ 250 the number of establishments with 250 or more employees.
(1)$$x_{ri}\;=\;\mathrm{employment}\;\mathrm{for}\;\mathrm{the}\;\mathrm{local}\;\mathrm{authority}\;r\;\mathrm{and}\;\mathrm{the}\;\mathrm{sector}\;\mathrm{of}\;\mathrm{activity}\;i$$
(2)$$x_{i}=\sum_{r=1}^{R}x_{ri}\;=\;\mathrm{employment}\;\mathrm{by}\;\mathrm{NUTS}\;\mathrm{II}\;\mathrm{for}\;\mathrm{the}\;\mathrm{sector}\;i$$
(3)$$x_{r}=\sum_{i=1}^{I}x_{ri}\;=\;\mathrm{employment}\;\mathrm{for}\;\mathrm{the}\;\mathrm{local}\;\mathrm{authority}\;r\;\mathrm{in}\;\mathrm{all}\;\mathrm{sectors}$$
(4)$$\sum_{r=1}^{R}\sum_{i=1}^{I}x_{ri}\;=\;\mathrm{employment}\;\mathrm{registered}\;\mathrm{by}\;\mathrm{NUTS}\;\mathrm{II},\;\mathrm{in}\;\mathrm{all}\;\mathrm{sectors}\;\mathrm{of}\;\mathrm{activity}$$
(5)$$x_{rj}\;=\;\mathrm{Employment}\;\mathrm{for}\;\mathrm{the}\;\mathrm{local}\;\mathrm{authority}\;r\;\mathrm{and}\;\mathrm{in}\;\mathrm{manufacturing}\;\mathrm{industry}$$
(6)$$PE_{R}\;=\;\mathrm{Population}\;\mathrm{employed}\;\mathrm{by}\;\mathrm{NUTS}\;\mathrm{II}$$
(7)$$A_{r}\;=\;\mathrm{Local}\;\mathrm{authority}\;\mathrm{area}\;r,\;\mathrm{in}\;{\mathrm{Km}}^{2}$$
(8)$$E\leq 9_{r}\;=\;Nº\;\mathrm{of}\;\mathrm{establishments}\;\mathrm{with}\;\mathrm{up}\;\mathrm{to}\;9\;\mathrm{employees}\;\mathrm{in}\;\mathrm{the}\;\mathrm{local}\;\mathrm{authority}\;r$$
(9)$$E\leq 49_{r}\;=\;Nº\;\mathrm{of}\;\mathrm{establishments}\;\mathrm{with}\;\mathrm{up}\;\mathrm{to}\;49\;\mathrm{employees}\;\mathrm{in}\;\mathrm{the}\;\mathrm{local}\;\mathrm{authority}\;r$$
(10)$$E\leq 249_{r}\;=\;Nº\;\mathrm{of}\;\mathrm{establishments}\;\mathrm{with}\;\mathrm{up}\;\mathrm{to}\;249\;\mathrm{employees}\;\mathrm{in}\;\mathrm{the}\;\mathrm{local}\;\mathrm{authority}\;r$$
(11)$$E\geq 250_{r}\;=\;Nº\;\mathrm{of}\;\mathrm{establishments}\;\mathrm{with}\;250\;\mathrm{or}\;\mathrm{more}\;\mathrm{employees}\;\mathrm{in}\;\mathrm{the}\;\mathrm{local}\;\mathrm{authority}\;r$$
(12)i=1, …,3
(13)r=1, …,308

Table 2, below, from expressions (1) to (13), presents the indicators of localization and specialization. The indicators of localization and specialization calculated are the following: Quotient of Localization $\left(QL_{ri}\right)$; Coefficient of Specialization $\left(CE_{r}\right)$; Rate of Industrialization $\left(TI_{r}\right)$; Density of Establishments by Business Size ($Micro_{r}$, $Small_{r}$, $Medium_{r}$,$Large_{r}$); and the Theil Index included in the indices of generalized entropy $\left(IT_{r}\right)$.

The localization quotient $\left(QL_{ri}\right)$ is the most widely used indicator in the literature, being explicitly recommended by Isard [110] to measure the relative level of concentration of the sector of activity *i* in local authority *r* and identify the relative centers of localization and specialization of activity *i* in the national territory. The indicator takes positive or null values and will be higher the higher the concentration of activity *i* in local authority *r*. The indicator has the value of 0 when sector *i* is not present in the local authority *r*; if the value is under 1, the weight of sector *i* in the local authority is relatively less than that of the area of reference. For values of 1, the relative importance of sector *i* in the local authority *r* is the same as the relative importance of the sector nationally, i.e., the regional and national concentration of sector *i* are identical. When the value of the indicator is above 1, this means that sector *i* is relatively concentrated in local authority *r*. A low value of the localization quotient reflects the absence of regional competitive advantage in that sector or simply lost opportunities [110]. The degree of regional specialization is analyzed through calculating the specialization coefficient $\left(CE_{r}\right)$, with the $\left(CE_{r}\right)$ of local authority *r* being a relative measure of the degree of regional specialization, which compares the regional sector structure with the sector structure of the area of reference. The indicator takes a null value (extreme situation), when the regional sector structure coincides with that of the area of reference. In this case, the local authority is not considered specialized. The closer to 1 the value of the indicator, the greater the distancing from the regional sector structure from that of the country, with the local authority being considered specialized. This indicator has the great advantage of summarizing in a single value the degree of relative specialization, compared to the localization coefficient, with the disadvantage of not indicating the sectors in which the region is specialized, but this failing can be overcome through complementary analysis of the localization quotient. The rate of industrialization $\left(TI_{r}\right)$ measures the percentage of the population employed in the manufacturing industry in relation to the total population employed in the local authority [113], and it can have positive or null values, being higher the greater the local authority’s industrialization. The density of establishments by business size ($Micro_{r}$, $Small_{r}$, $Medium_{r}$,$\;Large_{r}$) measures the number of establishments according to business size by km^2^. Employer size is according to the European classification, i.e., up to 10 employees refers to micro-firms, while small firms have between 10 and 49 employees, medium-sized ones have between 50 and 249 employees, and large ones have 250 or more employees. The Theil Index $\left(IT_{r}\right)$ is a compound index that can measure a local authority’s degree of specialization/diversification, as explained in the following section, which gives a brief description of the origin of the concept of entropy and the indices derived from generalized entropy.

#### Entropy and Its Measures

The concept of entropy, which in general terms is a measure of the dispersion of material in a given space, was developed and applied to a variety of subjects, including thermodynamics [114], kinetic theory [115], classic statistical mechanics [116], quantum statistical mechanics [117], and the theory of information [118]. Derived from information theory, measures of generalized entropy serve to measure the distribution of wealth, and according to Mussard et al. [119], different metrics such as the Herfindahl–Hirschman Index (HHI), Atkinson Index, Gini Index, and Theil Index, are particular cases of the class of measures of generalized entropy.

The Herfindhal Index, known as the Herfindahl–Hirschman Index, or HHI, which was proposed independently by Hirschman [120] and Herfindahl [121], includes the family of generalized indices of entropy. Later, Hirschman [122] claimed authorship of the index. HHI measures the concentration of industry using the data of all companies in a given industry and is written as follows:(14)$$\mathrm{HHI}\;=\;\sum_{i=1}^{N}\;S_{i}^{2}$$
where *N* is the number of companies; and $S_{i}$ is the market quota of company *i* in the market. The index varies from 1/*N* (lower limit) to 1 (upper limit)

The Atkinson index, also known as the Atkinson measure or Atkinson’s measure of inequality, proposed by Atkinson [123], represents the percentage of total income that a given group should have to give up for more partitions of equal income to be viable. The index varies from 0 (perfect equality) to 1 (maximum inequality) and is represented by:(15)$$A_{\varepsilon }=\;1\;-{\left[\frac{1}{n}\sum_{i=1}^{n}{\left(\frac{y_{i}}{\bar{y}}\right)}^{1-\varepsilon }\right]}^{\frac{1}{\left(1-\varepsilon \right)}}$$
where $\bar{y}$ is average income, $y_{i}$ is individual income, *i* is the number of individuals or families, and *ε* indicates the degree of aversion to disparity.

The Gini Index, also known as the Gini coefficient, was developed by Gini [124] to express inequality of wealth and is based on the Lorenz curve, which is the curve of accumulated frequencies that compares the distribution of income with uniform distribution representing equality. Application of the Gini coefficient in measuring inequalities can be limited to a part of the distribution, in this case, the part corresponding to the lower or upper extreme of income distribution. Considering $x_{i}$ as a point on the axis of *x* (representing the accumulated percentage of the population) and $y_{i}$ as a point on the axis of *y* (axis of the accumulated percentage of income), the Gini coefficient can be expressed as follows:(16)$$\mathrm{Gini}=\;1-\sum_{i=1}^{N}\left(x_{i}-x_{i-1}\right)\left(y_{i}+y_{i+1}\right).$$

When there are equal intervals on the axis of *x*, this is simplified
(17)$$\mathrm{Gini}\;=\;1-\frac{1}{N}\sum_{i=1}^{N}\left(y_{i}+y_{i+1}\right).$$

The Theil Index, proposed by Theil [125], belongs to the family of generalized indices of entropy. This measure serves fundamentally to analyze the distribution of wealth, and in this article, the Theil notion of entropy serves to analyze the diversity of sectors of economic activity present in a local authority, meaning here a measure of diversification of a given local authority. The degree of specialization/diversification obtained through the Theil index $\left(IT_{r}\right)$ depends only on the sectoral structure of the local authority analyzed. The limits of this indicator vary between 0 (indicating situations of maximum specialization) and the logarithm of the number of sectors of activity retained for analysis (signaling situations of total diversification). The result of the Theil Index $\left(IT_{r}\right)$ can also be normalized to vary between 0 and 1, representing maximum diversification and maximum specialization, respectively.

It is interesting to remember that the choice of index to use is directly related to the specific aspect to be studied. It is emphasized that use of the Theil Index is important, since it allows assessment of the specific structure of a region (or local authority), immediately classifying the position of regions (or local authorities). Using it for only one region (or local authority) diminishes the capacity to interpret the results, and so it is advantageous to analyze the results of this index in comparison with other regions (or other local authorities) presenting a relevant common reference framework such as geographical proximity, location, spatial concentration of related companies, similar development strategies, etc.

### 3.2. Threshold Regression Method

The regression model with the threshold effect, originally proposed by Tong [126] and Tong and Lim [126], emerged applied to the context of time series, allowing individual observations to be divided in regimes based on the value of an observed variable. This model divides the sample in classes based on the value of an observed variable, irrespective of exceeding any limit. Later, Hansen [127] introduced appropriate techniques for the threshold regression with panel data. Allowing fixed individual effects, the regression model with the threshold effect with panel data divides the observations in two or more regimes, depending on whether a threshold variable is below or above a threshold value and whether those regimes are distinguished by different regression slopes. Therefore, from panel data, the equation of the model of the single threshold type is expressed by the following equation:(18)$$y_{it}\;=\;µ+X_{it}\left(q_{it}<\gamma \right)\beta_{1}+X_{it}\left(q_{it}\geq \gamma \right)\beta_{2}\;+u_{i}+e_{it}$$
where i=1…N; t=1…T; $y_{it}$ is a scalar-dependent variable; $X_{it}$ is a regression vector; $q_{it}$ is a scalar threshold variable; and γ is the threshold parameter that divides the equation in two regimes with coefficients $\beta_{1}$ and $\beta_{2}$. In addition, the parameter *u_i_* corresponds to the individual effect and $e_{it}$ corresponds to the error term. With γ given, the ordinary least squares estimator of β is expressed by the following equation:(19)$$\hat{\beta }=\left\{X^{*}{\left(\gamma \right)}^{\prime }X^{*}\left(\gamma \right)\right\}{}^{-1}\;\left\{X^{*}{\left(\gamma \;\right)}^{\prime }\;y^{*}\right\}$$
where $X^{*}$ and $y^{*}$ belong to the group of deviations. The residual sum of squares is equal to $\hat{{e*}^{\prime }}$ and $\hat{e*\;}$. To estimate γ, a search can be made of the subset of the threshold variable $q_{it}$, since instead of the search being made for the whole sample, this can be restricted to the interval $\left(\bar{\gamma },\;\underline{\gamma }\right)$, which are quantiles of $q_{it}$. When γ is known, the model is not different from a common linear model, but in the case of γ being unknown, there is a parameter problem, which makes distribution of the estimator γ outside the standard. Given the above, Hansen [127] proved that $\hat{\gamma }$ is a consistent estimator for γ, arguing that the most suitable way to test γ = $\gamma_{0}$ is respecting a confidence interval using the method of non-rejection of the region with a likelihood ratio statistic expressed as follows:(20)$$LR_{1}\left(\gamma \right)=\left\{\frac{LR_{1}\left(\gamma \right)-LR_{1}\left(\hat{\gamma }\right)}{\hat{{\hat{\sigma }}^{2}}}\right\}\overset{P_{r}}{\to }\xi P_{r}\left(x<\xi \right)=\;{\left(1-\;e^{\frac{-x}{2}}\right)}^{2}.$$

Therefore, for a level of significance α, the lower limit corresponds to the maximum value, which is less than the quantile α, and the upper limit corresponds to the minimum value, which is less than the quantile α. For example, for a α = 0.1, 0.05 and 0.01, the quantiles are 6.53, 7.35, and 10.59, respectively. If $LR_{1}\left(\gamma_{0}\right)$ is more than c (α), then *H*_0_ is rejected. In turn, the test for the threshold effect is identical to the one used to test whether the coefficients are the same in each regime. The null hypothesis (*H*_0_) and the alternative hypothesis (*H_a_*) are expressed as:(21)$$H_{0}:\beta_{1}\;=\;\beta_{2}H_{a}:\beta_{1}\neq \beta_{2}.$$

The *F* statistic is given by:(22)$$F_{1}=\frac{\left(S_{0\;-\;S_{1}}\right)}{\hat{{\hat{\sigma }}^{2}}}$$
where *S*_0_ is the sum of squared errors obtained through estimating Equation (18) with the null hypothesis of the non-existence of any threshold; *S*_1_ is the sum of squared errors obtained from estimating the equation of the single threshold model of panel data (see Equation (18)); and ${\hat{\sigma }}^{2}$ is the residual variance of the regression of the single threshold model of panel data (see Equation (18)). As in *H*_0_, the threshold γ is not identified, and $F_{1}$ has a non-standard asymptotic distribution, Hansen [128] suggests using a bootstrap procedure for the critical values of the *F* statistic in order to test the significance of the threshold effect. If there are multiple thresholds, i.e., various regimes, Hansen [127] suggests estimation of a double threshold model, which can be expressed as follows:(23)$$y_{it}=\;µ+X_{it}\left(q_{it}<\gamma_{1}\right)\beta_{1}+X_{it}\left(\gamma_{1}\leq q_{it}<\gamma_{2}\right)\beta_{2}\;+X_{it}\left(q_{it}\geq \gamma_{2}\right)\beta_{3}\;+u_{i}+e_{it}$$
where $\gamma_{1}$ and $\gamma_{2\;}$ are the thresholds that divide the equation in three regimes with coefficients $\beta_{1}$, $\beta_{2}$, and $\beta_{3}$. The general approach of the threshold model to test multiple thresholds is similar to what is performed in the case of the single threshold model, albeit with some differences. The first difference concerns the estimating procedure, which can be a three-stage sequenced estimate (when there are only three regimes) of two limiting parameters. Here, the first stage involves the same estimating procedure as presented for the single threshold model, which produces the first estimate ${\hat{y}}_{1}$. By fixing this threshold parameter, in the second stage, the second threshold parameter ${\hat{y}}_{2}^{r}$ is estimated, minimizing the sum of the squared errors of the equation (see Equation (23)). In the third and final stage, the first threshold parameter is re-estimated, keeping the second threshold parameter fixed. This sequential three-stage estimate results in the asymptotically efficient estimator of the threshold parameters, ${\hat{y}}_{1}^{r}$ and ${\hat{y}}_{2}^{r}$. It is noted that these estimators have the same asymptotic distributions as the threshold estimate obtained from a single threshold model, which means that confidence intervals should be considered, similarly to what was mentioned above. The second difference concerns the inference about the threshold estimates. When the null of no threshold is rejected with the *F*_1_ statistic, it is necessary to make an additional test to discriminate between one and two thresholds. This test is carried out through application of a bootstrapping procedure, but now simulating the distribution of the *F*_2_ statistic, which is expressed as follows:(24)$$F_{2}=\frac{\left\{S_{1}\left({\bar{\hat{y}}}_{1}\right)-S_{2}^{r}\left({\bar{\hat{y}}}_{2}^{r}\right)\right\}}{{\hat{\sigma }}_{22}^{2}}$$
where *S*_1_ is the sum of the squared errors obtained from the estimate in the first stage; ${\hat{y}}_{2}^{r}$ is the sum of the squared errors obtained from the estimate in the second stage; and ${\hat{\sigma }}_{22}^{2}$ is the residual variance of the estimate in the second stage.

Variables and Specification of the Model

Using localization and specialization indicators, as well as other variables referring to the 308 Portuguese local authorities, for the period 2013–2017, a balanced panel was constructed. Table 3 presents the variables used and the corresponding description.

In this study, the dependent variable corresponds to the logarithmic transformation of productivity ($Logproduct_{r}$). The independent variables used are associated with the research hypotheses raised: rate of industrialization ($TI_{r}$); squared rate of industrialization ($TI_{r}{}^{2}$); cubic rate of industrialization ($TI_{r}{}^{3}$); diversification measured by the Theil Index ($IT_{r}$); and the term of interaction between exports and the Theil Index ($Export_{r}*\;IT_{r}$). Concerning the control variables, these are the number of higher education institutions ($IES_{r}$) and a variation in the number of clusters (${\mathrm{Clusters}}_{r}$) that can influence productivity in some way.

According to Conceição and Heitor [129], the low level of productivity in Portugal may be justified partly by the structure of the economy, which has a relatively high quota of non-specialized workers in sectors with intensive incorporation of the work production factor. This low level of education of the majority of the population is one of the main reasons why many companies continue in activities of low productivity and do not adopt more new technology [130]. Consequently, there is growing recognition that a more educated population can generally be more innovative and more able to absorb technological changes [130]. Therefore, HEIs create knowledge that they supply to the economy, leading to increased productivity and simultaneously to a better provision of human capital [131]. Regarding clusters, according to Porter [63], these can increase productivity in various ways, namely through better access to inputs and specialized workers. Porter [63] mentions that clusters typically allow better access to institutions, public goods, and infrastructure; provide greater incentives to achieve high productivity; and make it easier for companies to measure the performance of internal activities. Various empirical studies reveal the positive effect of clusters on productivity, as in the case of the study by Martin et al. [132], which shows that French companies benefit from localization externalities that increase productivity, and the study by Cainelli et al. [99] exploring the impact of agglomeration and diversity on total factor productivity, suggesting that clusters have a significant effect on companies’ total factor productivity.

In this empirical approach, the type of model used also allows definition of a threshold variable concerning specialization ($CE_{r}$), as well as variables in a dependent regime that correspond to the density of firms by business size: ($Micro_{r}$); ($Small_{r}$); ($Medium_{r}$); and ($Large_{r}$). Therefore, the threshold model is adopted to estimate the level of the specialization threshold ($CE_{r}$) and analyze its influence on the logarithmic transformation of productivity ($Logproduct_{r}$). The specification of the econometric model, with indication of the threshold equation regression, is given by the equation:(25)$$Logprodut_{r_{it}}=\;µ+TI_{r_{it}}+TI_{r_{it}}{}^{2}+TI_{r_{it}}{}^{3}+IT_{r}{}_{it}+Export_{r}*IT_{r_{it}}+IES_{r_{it}}+\;Clusters_{r_{it}}+X_{it}\left(CE_{r_{it}}{}_{\;}<\gamma_{1}\;\right)\beta_{1}+\;X_{it}\left(\gamma_{1}\leq CE_{r_{it}}{}_{\;}<\gamma_{2}\;\right)\beta_{2}\;+X_{it}\left(CE_{r_{it}}{}_{\;}\geq \gamma_{2}\;\right)\beta_{3}\;+u_{i}+e_{it}$$
where i=1,…,308; t=2013,…,2017; µ= Constant; $X_{it}=$ Regression vector (Micro, Small, Medium and Large); $\gamma_{1\;}$ and $\gamma_{2\;}=$ Threshold parameters that divide the equation; β= Coefficients; *u_i_* = Individual effect; and $e_{it}$= Error term.

## 4. Results

A double threshold model was tested, using a bootstrap method with 300 replications. First, for the single threshold model (Th-1), the results indicate that the estimator is 0.348 with a confidence interval of 95% [0.343; 0.352] (see Table 4).

Furthermore, the results show that in the test for the single threshold model (with $H_{0}$: linear model; $H_{1}$: single threshold model), the $F_{1}$ statistic of 78.040 is greater than its critical value of 64.294 for a level of significance of 1% (see Table 5). Therefore, the $F_{1}$ statistic is significant with a bootstrap *p* value of 0.003, indicating that $H_{0}$ is rejected. In other words, the relation between specialization ($CE_{r}$) and productivity ($Logproduct_{r}$) is not linear, and there is the threshold effect. For the double threshold model, the $F_{2}$ statistic (with *H*_0_: single threshold model; *H*_1_: double threshold model) is highly significant with a bootstrap *p* value of 0.000 ($F_{2}$ = 48.570 > Crit 1 = 38.295). This results in rejecting *H*_0_, suggesting the detection of a double threshold model with the estimates of 0.348 (Th-21) and 0.023 (Th-22) (see Table 4).

The results of the fixed effects regression and threshold effect are presented in Table 6.

The F statistic of 3.06, for a level of significance of 5% with the null hypothesis of all $u_{i}$=0 confirms that the fixed effect model is appropriate. Considering that the model is appropriate and that the regression estimates in the double threshold model indicate the effect of specialization in three regimes, the summarized description of the most significant variables found follows, with subsequent reference to the tests of the research hypotheses.

The parameters displayed in Table 6 for the variables $TI_{r}$, $TI_{r}{}^{2}$, $TI_{r}{}^{3}$, $IT_{r}$, $Export_{r}*IT_{r}$, $IES_{r}$, and $Clusters_{r}$ have been estimated through panel data regression with fixed effects, as well as considering the admission of the threshold effect, which provides the calculation of the estimators relating to business size: $Micro_{r}$, $Small_{r}$, $Medium_{r}$, and $Large_{r}$, which vary according to the three specialization regimes, in corresponding terms.

It is worth pointing out that the rate of industrialization ($TI_{r}$), cubic transformation of the rate of industrialization ($TI_{r}{}^{3}$), the Theil Index representing diversification ($IT_{r})$, the term of interaction between the weight of exports and the Theil Index ($Export_{r}*IT_{r}$), and higher education institutions ($IES_{r}$) have a positive and significant (1%) influence on productivity ($Logproduct_{r}$). In addition, with a significance of 1%, the squared transformation of the rate of industrialization ($TI_{r}{}^{2}$) and the variation in the number of clusters ($Clusters_{r}$) have a negative influence on productivity ($Logproduct_{r}$).

In addition, when considering the increased rate of industrialization, performing, firstly, a squared transformation of the rate of industrialization and then calculating the first order partial derivative given by $\frac{\partial Logproductr}{\partial TI_{r}}=0.4234-0.0884x+0.0039x^{2}=0$, it is possible to identify 6.874 as the maximum and 15.792 as the minimum rate of industrialization. Explicitly, *x* = $\frac{-\left(-0.0084\right)-\surd \left(-0.0884^{2}\right)-4\times 0.0039\times 0.4234}{2\times 0.0039}=6.874$ and *x* = $\frac{-\left(-0.0084\right)+\surd \left(-0.0884^{2}\right)-4\times 0.0039\times 0.4234}{2\times 0.0039}=15.792$, which correspond to red points in Figure 2.

The result obtained implies that for a given rate of industrialization up to 6.874, industrialization is a determinant factor restricting productivity, while above this figure, increased industrialization is found to be a determinant factor with a positive effect on productivity. Consequently, the empirical results obtained now indicate the existence of a non-linear relation between the rate of industrialization and productivity. In turn, calculation of the second-order partial derivative given by $\frac{\partial Logproductr^{2}}{\partial TI_{r}{}^{2}}=-0.0884\;+\;0.0078\;x=0$ identifies 11.333 as the point of inflection from which industrialization stimulates productivity. In more detail, *x* = $\;\frac{0.00884}{0.0078}=11.333$, which corresponds to the yellow point in Figure 2.

When the analysis is made considering the density of establishments by business size, for a specialization ($CE_{r}$) < 0.023, a positive coefficient of 0.003 is detected, which implies a significant (1%) and positive relation between micro-firms ($Micro_{r}$) and productivity ($Logproduct_{r}$); when the specialization is 0.023 $\leq CE_{r}<$ 0.348, the positive coefficient of 0.000 suggests a positive but non-significant relation between micro-firms ($Micro_{r}$) and productivity ($Logproduct_{r}$), and when specialization ($CE_{r}$) is ≥ 0.348, the negative coefficient of −0.042 suggests that micro-firms ($Micro_{r}$) cause a negative and significant (1%) effect on productivity ($Logproduct_{r}$). Concerning small firms ($Small_{r}$), only for a specialization ($CE_{r}$) < 0.023 do we find a negative and significant (1%) effect on productivity ($Logproduct_{r}$). For higher levels of specialization for small firms ($Small_{r}$), there is no significant effect on productivity ($Logproduct_{r}$). As for medium-sized firms ($Medium_{r}$), when the specialization ($CE_{r}$) is <0.023, they produce a positive and significant (1%) effect on productivity, the same occurring for a specialization ($CE_{r}$) ≥ 0.348 but for a level of significance of 10%. For a specialization 0.023 $\leq CE_{r}<$ 0.348, a negative and significant (1%) effect on productivity ($Logproduct_{r}$) is found. Concerning large firms ($Large_{r}$), only for levels of specialization ($CE_{r})\;<$ 0.023 and ≥0.023 do we find a negative and significant (1%) effect on productivity. The results obtained when considering the density of establishments according to different business size—micro, small, medium, and large—are explained by the Portuguese business sector being formed mainly of micro and small firms, and it should be underlined that firms of this size show structural shortcomings regarding the quality of management and organization of processes and production that tend to improve productivity.

## 5. Discussion

Considering Hypothesis 1, proposing a positive relationship between industrialization and productivity, this cannot be rejected, since a significant and positive effect on productivity is found. This result agrees with the study by Rodrick [66], which found, using a large sample of countries, that industry is the only sector of the economy that achieves unconditional convergence in productivity. More evidence is obtained in previous studies—for example, McCausland and Theodossiou [67], Kathuria and Natarajan [68], Güçlü [69], and Szirmai and Verspagen [70], where it is concluded that the industrial sector causes a positive impact on economic growth, underlining in this connection the perspectives of Kathuria and Natarajan [68] and Szirmai and Verspagen [70], according to whom the role of the service sectors is not comparable to that of the industrial sector.

Hypothesis $H_{2}$ proposing a non-linear relation between an increased rate of industrialization and productivity is not rejected. Therefore, for squared transformation of the rate of industrialization, its effect is significant, but it has a negative impact on productivity. On the other hand, for cubic transformation of the rate of industrialization, its impact is significant and positive. Although there is no previous empirical evidence to directly corroborate the non-linear relationship between industrialization and productivity, considering the set of evidence obtained previously by Ortiz et al. [73], taking economic growth as the dependent variable, the non-linear relation between economic growth and industrialization is confirmed, which is in line with these authors’ argument indicating that each society should strive to achieve a minimum level of industrial technological integration before being able to reap the benefits of economic growth arising from industrialization.

Concerning Hypothesis $H_{3}$, proposing that diversification is positively related to productivity, a positive and significant influence is found, meaning the hypothesis is not rejected. In this connection, Jacobs [79] argues that a diversification of industries in one place promotes knowledge-related externalities and leads to innovation and economic growth. Added to this is the view of Batisse [100], according to whom specialization has a strong negative impact on growth, contrasting with the positive impact associated with a more diversified industrial basis.

Regarding Hypothesis $H_{4}$ aiming to test the hypothetical moderating effect of exports on the relation between diversification and productivity, a positive, significant effect on productivity is found. This evidence agrees with the results of previous empirical studies by Aw and Hwang [101], Bernard and Wagner [102], Bernard and Jensen [103], Aw et al. [104], and Delgado et al. [105], which indicate that exporting companies will have better performance than non-exporting ones, not only in terms of survival, salaries, capital intensity, and technological sophistication, but also in productivity. In addition, regarding exports affecting the relation between diversification and productivity, Prebisch [106] claims that diversification of the productive structure is beneficial for economic growth by making the country less dependent on more sophisticated imports, and therefore reducing the tendency toward external imbalance and a low level of economic growth in such economies.

## 6. Conclusions

The empirical evidence obtained indicates a non-rejection of the research hypotheses, and so industrialization is revealed to have a significant and positive relation with productivity. There is also evidence of non-linearity in the relation between increased industrialization and productivity, considering the line previously proposed by Ortiz et al. [73], which indicates the need to intensify industrial technological integration before being able to reap the benefits of industrialization for productivity, corresponding to increased economic growth. The moderating effect of exports on the relation between diversification and productivity is also found to be positive and significant, corroborating the existence of an acceleration effect of the “competitive productivity kit” type on the positive relation between the rate of industrialization and productivity. HEIs are also seen to have a positive and significant effect on productivity, and in this connection, it is true that HEIs create knowledge that they supply the economy with, leading to increased productivity and simultaneously a better provision of qualified human capital [131]. The control variable relating to the variation of clusters is also significant, but its impact on productivity is negative, which warrants continuous reflection by political decision-makers, planners, business people, and higher education institutions, toward the design and implementation of new practices and policies to strengthen the industrial critical mass, with the ultimate aim of raising productivity. Considering different regions of specialization by business size, the results obtained reveal that when it is a question of micro-firms and low levels of specialization, the impact of productivity is positive and significant, but for higher levels of specialization, the impact is negative. Small companies have a negative and significant effect on productivity when specialization is low. For high levels of specialization, small companies are found not to cause an impact on productivity. For a very low level of specialization, medium-sized firms have a positive and significant impact on productivity, and for an intermediate level of specialization, their impact on productivity is negative and significant. For higher and lower regimes of specialization, large companies have a negative and significant effect on productivity, which is not unrelated to the persistent gaps in terms of management quality and the need to update productive structures with greater technological intensity. This diversity of results according to business size is justified by the weight of micro, small, and medium-sized firms, accounting for around 99% of the Portuguese business sector. It is important to note that micro, small, and medium-sized companies differ from large ones in various ways [133]. For example, they have limited resources in terms of management, workforce, and finance [134], and they seem to be more flexible and accompanied by less formalized processes than in large firms, which can facilitate innovation.

One of the main limitations of this study arises from the unavailability of disaggregated data at the local authority level. For example, the rate of industrialization was calculated by considering the employed population in each NUTS II, according to data available in the 2011 census. Another limitation concerns the shortage of empirical studies on the relation between industrialization and productivity, it being more usual to analyze economic growth as a dependent variable, although productivity can be considered as a factor stimulating economic growth or even a proxy to measure the performance of the unit analyzed.

This study gives rise to a number of implications. In the first place, considering the results obtained, which aim to study the Portuguese productive structure at the local authority level, it is suggested that political decision-makers should somehow encourage regions to increase the industrial critical mass, as well as the diversity of their productive activity through clusters, as by doing so they will contribute to regions becoming more resilient and competitive, given the major opening up of the Portuguese economy and the likely impacts of external shocks. The creation of new instruments is also suggested, aiming for a more vertical structure of industry that is directed complementally toward crossed fertilization between different stages of the chain and between different industries, which should also be based on seeking stronger connections of horizontal integration within industries themselves and in establishing open innovation relations with universities, incubation structures, laboratories, and research units.

Finally, considering the urgent need to strengthen the diversification of productive activities, one suggestion for future research is extending this study to the level of European NUTS II and NUTS III regions for the better mapping of sectors of economic activity and Key-Enabling Technologies (KETs), which can contribute to reinforcing productivity and competitiveness, following a sustainable logic of strategic diversification of sectors of economic activity and considering the spatial heterogeneity of the European regional chessboard.

## Figures and Tables

**Figure 1 entropy-22-01271-f001:**
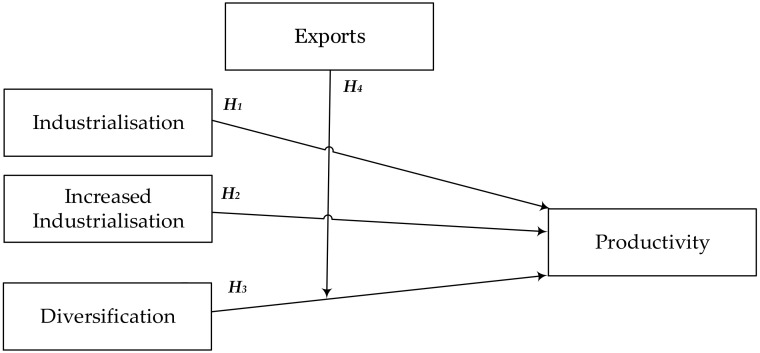
Determinants of concentration and specialization of productivity: Model and hypotheses. **Source:** Own elaboration.

**Figure 2 entropy-22-01271-f002:**
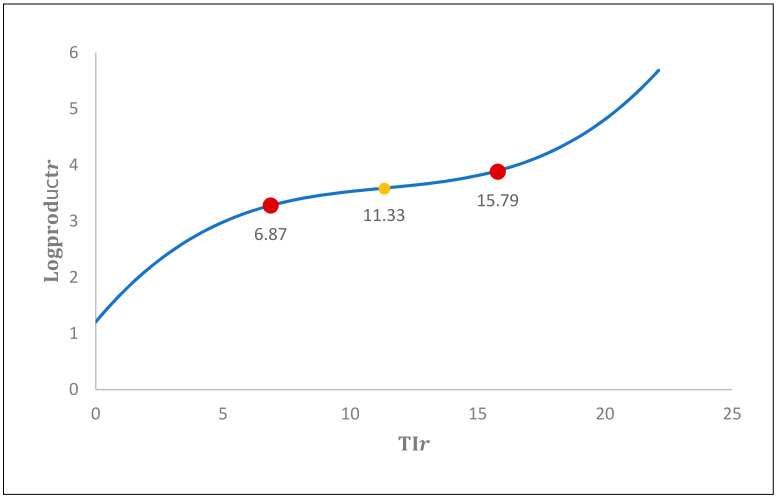
Relation between industrialization and productivity. **Source**: Own elaboration.

**Table 1 entropy-22-01271-t001:** Portuguese Classification of Economic Activities: CAE-Rev.3.

CAE-Rev.3	
Section	Designation
A	Agriculture, livestock, hunting, forestry and fishing
B	Mining industry
C	Manufacturing industry
D	Electricity, gas, steam, hot and cold water and cold air
E	Water storage, treatment and distribution; sanitation, waste management and depollution
F	Construction
G	Wholesale and retail commerce; car and motorcycle repair
H	Transport and storage
I	Accommodation, restaurants, and similar
J	Information and communication activities
K	Financial and business activities
L	Real estate activities
M	Consultancy, scientific, technical and similar activities
N	Administrative activities and support services
O	Public administration and defense; obligatory social security
P	Education
Q	Human health activities and social support
R	Artistic, performance, sporting, and recreational activities
S	Other service activities
T	Activities of families employing domestic staff and family production activities for own consumption
U	Activities of international bodies and other foreign institutions

**Source:** Adapted from the National Statistics Institute (INE). https://www.ine.pt/ine_novidades/semin/cae/CAE_REV_3.pdf.

**Table 2 entropy-22-01271-t002:** Indicators of Localization and Specialization.

Indicators	Metrics	Reference
Quotient of Localization ($QL_{ri}$)	$QL_{ri}$=$\frac{\left(\frac{x_{ri}}{x_{r}}\right)}{\left(\frac{x_{i}}{x}\right)}$, $QL_{ri\;\geq 0}$	$\left(\frac{\frac{Loc.Authority}{Loc.Authority}}{\frac{NutsII}{NutsII}}\right)$
Coefficient of Specialization $\left(CE_{r}\right)$	$CE_{r}$=$\frac{\sum_{i=1}^{I}\left|\frac{x_{ri}}{x_{r}}-\left.\frac{x_{i}}{x}\right|\right.}{2}$, $CE_{r\;\in \;\left[0,1\right]}$	$\left|\frac{Loc.Authority}{Loc.Authority}\right.-\left.\frac{NutsII}{NutsII}\right|$
Rate of Industrialization $\left(TI_{r}\right)$	$TI_{r=\frac{x_{rj}}{PE_{R}}}$	$\left(\frac{Loc.Authority}{NutsII}\right)$
Density of Establishments by Business Size ($Micro_{r}$, $Small_{r}$, $Medium_{r}Large_{r}$)	$Micro_{r=\;}\frac{E\leq 9_{r}}{A_{r}}$, $Micro_{r}\geq \;0$ $Small_{r=\;}\frac{E\leq 49_{r}}{A_{r}}$, $Small_{r}\geq \;0$ $Medium_{r=\;}\frac{E\leq 249_{r}}{A_{r}}$, $Medium_{r}\geq 0$ $Large_{r=\;}\frac{E\geq 250_{r}}{A_{r}}$, $Large_{r}\geq 0$	$\left(\frac{Loc.Authority}{Loc.Authority}\right)$
Theil Index $\left(IT_{r}\right)$	$IT_{r}$= − $\sum_{i=1}^{I}\;\left[\frac{xri}{xr}*log\left(\frac{xri}{xr}\right)\right]$,$\;0\leq IT_{r}$ ≤ log *I* Normalize Theil Index $\left(IT_{r}\right)$ $IT_{r}=\frac{logI-\;IT_{r}}{logI},\;0\leq IT_{r}\;\leq \;1$	$\left[\frac{Loc.Authority}{Loc.Authority}*\;log\left(\frac{Loc.Authority}{Loc.Authority}\right)\right]$

**Source:** Elaborated by the authors, based on Cerejeira [22] and Simões Lopes [112].

**Table 3 entropy-22-01271-t003:** List and description of variables.

	Variable	Description
Dependent variable	$Logproduct_{r}$	Logarithmic transformation of productivity
Independent variables	$TI_{r}$	Rate of industrialization
	$TI_{r}{}^{2}$	Squared rate of industrialization
	$TI_{r}{}^{3}$	Cubic rate of industrialization
	$IT_{r}$	Theil Index (diversification)
	$Export_{r}*\;IT_{r}$	Term of interaction between the weight of exports and the Theil Index
Control variables	$IES_{r}$	Number of higher education Institutions, by local authority
	$Clusters_{r}$	Variation in number of clusters ^1^
Threshold variable	$CE_{r}$	Coefficient of specialization
Variables in dependent regime	$Micro_{r}$	Density of micro-firms
	$Small_{r}$	Density of small firms
	$Medium_{r}$	Density of medium firms
	$Large_{r}$	Density of large firms

^1^ The variation in the number of clusters for 2013, 2014 and 2015 obtained from the sums of centres of competitiveness identified by the programme of Compete 2007 and 2013 and the clusters identified by the same programme for 2003–2015 less the clusters that had been identified by Porter in 1992 and for the years 2016 and 2017 the variation was obtained through the clusters of competitiveness recognised by IAPMEI in 2015 less the centres of competitiveness identified by the Compete programmes 2007 and 2013 less the clusters identified by the same programme for 2003–2015. **Source:** Own elaboration.

**Table 4 entropy-22-01271-t004:** Threshold estimator.

	Threshold	Confidence Interval for 95%
**Th-1**	0.348	[0.343;0.352]
**Th-21**	0.348	[0.343;0.352]
**Th-22**	0.023	[0.017;0.023]

**Source:** Own elaboration.

**Table 5 entropy-22-01271-t005:** Effect of the threshold test.

Test of the Threshold Effect (Bootstrapping; n.º of Replications = 300)							
Threshold	RSS	MSE	F	P	Crit10%	Crit5%	Crit1%
**Single**	235.237	0.191	78.040	0.003	46.118	51.730	64.294
**Double**	226.314	0.184	48.570	0.000	28.581	32.006	38.295

**Source:** Own elaboration.

**Table 6 entropy-22-01271-t006:** Regression estimates: double threshold model.

Variables	Fixed Effects	
	Coefficients	P > |t|
$Logproduct_{r}$		
$TI_{r}$	0.423	0.000 ***
$TI_{r}{}^{2}$	−0.044	0.000 ***
$TI_{r}{}^{3}$	0.001	0.000 ***
$IT_{r}$	0.881	0.000 ***
$Export_{r}*IT_{r}$	3.446	0.000 ***
$IES_{r}$	0.021	0.000 ***
$Clusters_{r}$	−0.019	0.005 ***
	**Threshold effect**	
	**Coefficients**	**P > |t|**
$Micro_{r}$		
$CE_{r}$*(*$CE_{r}<$ 0.023)	0.003	0.009 ***
$CE_{r}$ (0.023 $\leq CE_{r}<$ 0.348)	0.000	0.747
$CE_{r}$ ($CE_{r}\geq \;$ 0.348)	−0.042	0.000 ***
$Small_{r}$		
$CE_{r}$ ($CE_{r}<$ 0.023)	−0.167	0.001 ***
$CE_{r}$ (0.023 $\leq CE_{r}<$ 0.348)	0.002	0.891
$CE_{r}$ ($CE_{r}\geq \;$ 0.348)	0.415	0.219
$Medium_{r}$		
$CE_{r}$ ($CE_{r}<$ 0.023)	1.232	0.000 ***
$CE_{r}$ (0.023 $\leq CE_{r}<$ 0.348)	−0.263	0.000 ***
$CE_{r}$ ($CE_{r}\geq \;$ 0.348)	1.910	0.092 *
$Large_{r}$		
$CE_{r}$ ($CE_{r}<$ 0.023)	−2.786	0.000 ***
$CE_{r}$ (0.023 $\leq CE_{r}<$ 0.348)	0.063	0.828
$CE_{r}$ ($CE_{r}\geq \;$ 0.348)	−80.844	0.000 ***

**constant**	1.203	0.000
F test of all u_i = 0: F(4.152) = 3.06Prob < F = 0.016		

* significance 10%| *** significance 1%. **Source:** Own elaboration.

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
