# Peer review of "Spatial and Sectoral Determinants of Productivity: An Empirical Approach Using an Entropy Lens"

_entropy, 2020, doi:10.3390/e22111271_

Round 1
Reviewer 1 Report
Please see the attached file.

Author Response
Please find the answers in attachment.

Reviewer 2 Report
I propose to add literally one sentence in the Abstract explaining why the Authors:
1) selected the period 2013-17 for the study
2) decided to choose a threshold model.
This will make the work easier for the average reader to read.
Work, already at the Abstract level, is intriguing/inspiring, for example when it states that (I quote): "For large firms, the impact on productivity is negative for both high and low levels of specialization, ...". Work is an important stimulus /inspiration to conduct research into companies' contribution to productivity. The issue is complex because it requires the analysis of many coexisting dynamics.
The work is neatly composed - it is pleasant to read. The introduction and review of the literature is written in a balanced manner in the convention adopted in the world of economics - it is also acceptable to the world of econo- and sociophysics. The work gives the impression of a holistic approach to the subject, which can be very useful for an average reader. The authors convincingly point to the literature sources that inspired them and with which they compared their results.
The authors present the quantitative approach in Sec. 3. The authors properly selected the set of necessary indicators based on the literature. The work is extensive but interesting and convincing, so it is not too boring.
Subsec. 3.1.1 seems to be of particular interest to me. At the outset, I have a note regarding the markings: in eq. (14) it should become (I think) 'IHH 'and not' HH 'as it is now. The Authors clearly show the role of the \gamma threshold (Eq. (18)) in the one-threshold model and the \gamm1 and \gamma2 thresholds in the two-threshold model (Eq. (23)). The work is basically of a review and application character (at least I perceive it that way) - it may be difficult to distinguish what is new in the methodology about the Authors. The Authors should emphasize it somehow in the text of the work. By the way, eq. (24) requires technical improvement.
However, part of the Sec. 4 is suitable for the Appendix - here it would be enough to refer to it. I think that almost all math formulas should be moved to Appendix. This section should only contain words and formulas in sentences expressed with references to this new Appendix. In this context, Fig. 2 and its discussion is very important. However, I propose to specify explicitly which expressions the Authors use when determining the position of both the red points and the yellow point. By the way, in the formula (29) in the second line in the expression: "... - 0.0189) Clusters_ {r_ {it}} ..." the bracket ")" is redundant.
Especially important to me is the fact that (I quote): “… evidence of non-linearity in the relation between increased industrialization and productivity. I like the Sec. Conclusions saturated with important results and conclusions.
After introducing the corrections I indicated, I recommend the paper for publication in the Entropy journal.
Author Response
Please find the answers in attachment.

Round 2
Reviewer 1 Report
I have read the new version of the paper. I stress once again my main critical point, i.e. the concept of Entropy is just marginally employed in the empirical analysis. I highlight once again that “The paper is well explained and the empirical analysis is exhaustive.”, as I already written in my previous report. However, I think is more suited for a journal specialized in industrial dynamics.
Reviewer 2 Report
The authors carefully responded to my comments. The work is of an application nature and may serve as a useful guide for other researchers. I recommend the paper for publication in the entropy journal.